# Rewriting by Generating: Learn to Solve Large-Scale Vehicle Routing Problems

## Abstract

The large-scale vehicle routing problems are defined based on the classical VRPs with thousands of customers. It is of great importance to find an efficient and high-quality solution for real-world applications. However, existing algorithms for VRPs including non-learning heuristics and RL-based methods, only perform well on small-scale instances with usually no more than a hundred customers. They are unable to solve large-scale VRPs due to either high computation cost or explosive solution space that results in model divergence. Inspired by the classical idea of Divide-and-Conquer, we present a novel Rewriting-by-Generating(RBG) framework with hierarchical RL agents to solve large-scale VRPs. RBG consists of a rewriter agent that refines the customer division globally and an elementary generator to infer regional solutions locally. Extensive experiments demonstrate the effectiveness and efficiency of our proposed RBG framework. It outperforms LKH3, the state-of-the-art method for CVRPs, by $2.43\%$ when customer number $N = 2000$ and shortens the inference time by about 100 times[1].

## 1 Introduction

The *Large-Scale Vehicle Routing Problems* (VRPs) is an important combinatorial optimization problem defined upon an enormous distribution of customer nodes, usually more than a thousand. An efficient and high-quality solution to large-scale VRPs is critical to many real-world applications. Meanwhile, most existing works focus on finding near-optimal solutions with only no more than a hundred customers because of the computational complexity (Laporte, 1992; Golden et al., 2008; Braekers et al., 2016). Originated from the NP-hard nature as a VRPs, the exponential expansion of solution space makes it much more difficult than solving a small-scale one. Therefore, providing effective and efficient solutions for large-scale VRPs is a challenging problem (Fukasawa et al., 2006).

Current algorithms proposed for routing problems can be divided into traditional non-learning based heuristics and reinforcement learning (RL) based models. Many routing solvers involve heuristics as their core algorithms, for instance, ant colony optimization (Gambardella et al., 1999) and LKH3 (Helsgaun, 2017), which can find a near optimal solution by greedy exploration. However, they become inefficient when the problem scale extends. Apart from traditional heuristics, RL based VRPs solvers have been widely studied recently to find more efficient and effective solutions (Dai et al., 2017; Nazari et al., 2018; Bello et al., 2017; Kool et al., 2019; Chen & Tian, 2019; Lu et al., 2020). Thanks to the learning manner that takes every feedback from learning attempts as signals, RL based methods rely on few hand-crafted rules and thus can be widely used in different customer distributions without human intervention and expert knowledge. Besides, these RL methods benefit from a pre-training process allowing them to infer solutions for new instances much faster than traditional heuristics. However, current RL agents are still insufficient to learn a feasible policy and generate solutions directly on large-scale VRPs due to the vast solution space, which is usually $N!$ for $N$ customers. More specifically, the solution space of a large-scale VRPs with 1000 customers is $e^{2409}$ much larger than that of a small-scale one with only 100 customers. Consequently, the complexity makes the agent difficult to fully explore and makes the model hard to learn useful knowledge in large-scale VRPs.

To avoid the explosion of solution space in large-scale VRPs, we consider leveraging the classic Divide-and-Conquer idea to decompose the enormous scale of the original problem. In particularly,

---

[1]Codes and data will be released at https://github.com/RBG4VRPs/Rewriting-By-Generating

dividing the large-scale customer distributions into small-scale ones and then generating individual regional solutions to reduce the problem complexity. However, how to obtain a refined region division where the local VRPs can be handled effectively and how to coordinate iterations between global and local optimization efficiently remain two challenges of our VRPs solvers.

To tackle those two challenges above, we propose an RL-based framework, named **Rewriting-by-Generating (RBG)**, to solve large-scale VRPs. The framework adopts a hierarchical RL structure, which consists of a "Generator" and a "Rewriter". Firstly, We divide customers into regions and use an elementary RL-based VRPs solver to solve them locally, known as the "Generation" process. After that, from a global perspective, a special "Rewriting" process is designed based on all regional generations, which rewrites the previous solution with new divisions and the corresponding new regional VRPs results. Within each rewriting step, we select and merge two regions into a hyper-region, and then further divide it into two new sub-regions according to the hyper-regional VRPs solution. By doing this, the problem scale is decomposed into pieces and the problem could be solved efficiently using regional RL-based solvers, and can still preserve the solution quality which is improved by the rewriter continuously.

Extensive experiments demonstrate that our RBG framework achieves significant performance in a much more efficient manner. It has a significant advantage on solution quality to other RL-based methods, and outperforms the state-of-the-art LKH3 (Helsgaun, 2017), by $2.43\%$ with the problem size of $N = 2000$ and could infer solutions about $100$ times faster. Moreover, it also has a growing superiority to other methods when the problem scale increases.

**Notations:** We introduce some fundamental notations of large-scale VRPs, while the complete formulation is presented in the Appendix. Let $G(V, E)$ denote the entire graph of all customers and the depot. Specifically, $V = \{v_0, v_1, ..., v_i, ..., v_N\}$, where $v_0$ denotes the depot, and $v_i (1 \leq i \leq N)$ denotes the $i$-th customer with its location $(x_i, y_i)$ and its demand $d_i$. The edge $e_{i,j}$, or $E(v_i, v_j)$ in another manner represents the traveling distance between $v_i$ and $v_j$. Within the RBG framework, the generated regional VRPs solution $\pi_k = \{v_{k,0}, v_{k,1}, v_{k,2}, ..., v_{k,N_k}\}$ of the divided region $G_k$ has a corresponding traveling cost $C(\pi_k) = \sum_{i=0}^{N_k} E(v_{k,i}, v_{k,i+1})$. The entire solution of all customers is denoted by $\pi$.

## 2 RELATED WORK

We discuss previous works which are related to our research in the following two directions:

**Traditional Heuristics.** Since the exact methods (Laporte, 1992; Laporte & Nobert, 1987; Holland, 1992; Baldacci et al., 2010) are almost impossible to solve VRPs within a reasonable time due to the high computation complexity, researchers developed heuristics, i.e., non-exact methods, to find approximation solutions instead. Tabu search is one of the old metaheuristics (Glover, 1990b;a; Gendreau et al., 1994; Battiti & Tecchiolli, 1994), which keeps searching for new solutions in the neighborhood of the current solution. Instead of focusing on improving merely one solution, genetic algorithms operate in a series of solutions (Goldberg, 1989; Holland, 1992). It constructs new structures continuously based on parent structures. Instead of treating objectives to be optimized altogether, ant colony optimizations as another widely accepted solver, utilize several ant colonies to optimize different functions: the number of vehicles, the total distance and others(Gambardella et al., 1999; Dorigo et al., 2006; Dorigo & Di Caro, 1999). Meanwhile, recreate search methods keep constructing the current solution and ruining the current ones to build better solutions.(Schrimpf et al., 2000). This helps to expand the exploration space to prevent the local optimum. Among these categories, LKH3 is a state-of-the-art heuristic solver that empirically finds optimal solutions (Helsgaun, 2017).

Although these heuristics, compared to exact methods, can improve searching efficiency, they are still much too time-consuming when applied to large-scale VRPs with the acceptable performance required, and may fail to respond to any real-time solution requests.

**RL based VRPs Solutions.** Since the learning manner of reinforcement learning allows the agent model to directly infer solutions based on a pre-trained model with much shorter computation time, RL becomes a compelling direction on solving combinatorial optimizations. It has been successfully applied in VRPs particularly (Bello et al., 2017; Nazari et al., 2018; Kool et al., 2019). Vinyals et al. (2015) was the first to adopt deep learning in combinatorial optimizations by a novel *Pointer Network* model. Inspired by this, Bello et al. (2017) proposed to use RL to learn model parameters as an

optimal strategy instead of relying on ground-truth in a supervised learning way, which demonstrates the effectiveness on TSP and the knapsack problem. Nazari et al. (2018) further followed the idea to solve VRPs with attention mechanism as augmentation, and Kool et al. (2019) solved more generalized combinatorial optimization problems. Other than using the idea of $PointerNetwork$, Dai et al. (2017) develops their method over graphs via Q-learning (Sutton & Barto, 2018), so that the solutions could have better generalization ability. Chen & Tian (2019) proposed a local rewriting rule that keeps rewriting the local components of the current situation via a Q-Actor-Critic training process(Sutton & Barto, 2018). Lu et al. (2020) further developed a Learn-to-Iterate structure that not only improves the solution exploration but also generates perturbations to avoid local optimum. This is the first machine learning framework that outperforms LKH3 on CVRPs (capacitated VRPs), both in computation time and solution quality.

However, these existing RL based methods only achieve promising results without any hand-craft rules and expertise at small scales with usually no more than a hundred customers. The proposed models cannot be trained for thousand-customer-level VRPs because the state space and action space extend exponentially as the number of customers increases, and it will be hard for the model to learn useful route generation policy. In contrast, we propose an RL based framework formed upon the classical idea of Divide-and-Conquer to solve the large-scale challenge.

## 3 REWRITING-BY-GENERATING

Figure 1 shows the overview structure of our proposed framework, named Rewriting-by-Generating (RBG). Along with the fundamental idea of Divide-and-Conquer to decompose the enormous problem scale as discussed previously, we aim at dividing the total customers into separate regions and generate near-optimal solutions individually. To achieve this, we design a hierarchical RL structure including two agents which take different functions.

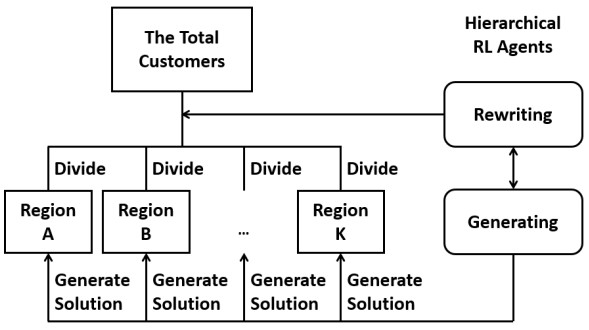

Figure 1: The overview of the Rewriting-by-Generating.

*First*, to refine and obtain more reasonable division results, we design the "Rewriting" process which keeps updating new divisions by rewriting the previous ones and their corresponding regional solutions. The division quality is critical to the final solution since customers from different regions cannot be scheduled upon the same route. Within each rewriting step, the agent selects and merges two regions based on their generated solutions. A new solution will be generated upon the merged hyper-region in the following step, and the rewriter will further divide the merge hyper-region back to two new regions. Since the exploration on different customer composition is complicated and it is not trivial to measure the direct influence to the final performance in terms of traveling distance, an RL-based rewriter is a wise choice to learn the selection and merging action. We will show that the model converges and achieves high performance when the rewriter agent learns a stable division result in Section 4.

*Second*, to reach the global solution from the regional scratches, we employ an elementary VRPs generator that generates solutions to each region, known as the "Generating" process. Considering the time efficiency and the ability to learn to solve certain customer distributions when the division updates continuously, we also apply an RL agent to learn to generate solutions on these smaller-scale regions.

*Overall*, we develop a hierarchical RL framework by coordinating the rewriter and the generator in two different scales iteratively. The rewriter updates new division and brings new customer distributions to the generator, while the solutions from the generator formulate a key component of the rewriter. From the technical perspective, it is worthy to note that the merging-repartitioning operation that our rewriter conducts is also adopted in previous meta-heuristics(Baker & Ayechew,

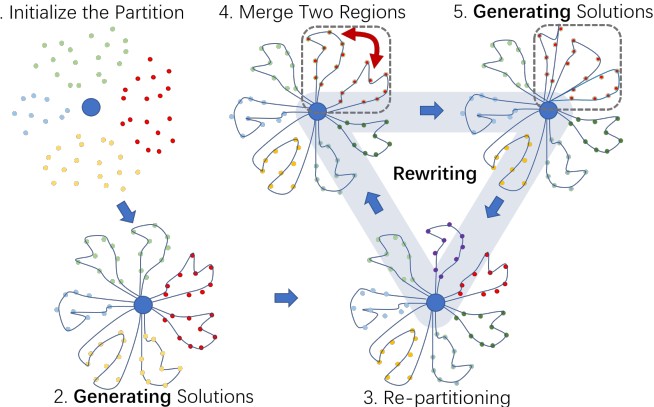

Figure 2: The core idea behind our Rewriting-by-Generating framework, where nodes denote customers with color representing regions, and solid lines denote routes.

2003; Bell & McMullen, 2004), while we replace the handcrafted heuristic with a learning agent. The global RL based rewriter is responsible for managing inter-regional exploration while the generator optimizes local results. The combination of Operation Research(OR) heuristics and RL guarantees an effective exploration process as well as achieving high computation efficiency from the prospect of fast solution generation on inference instances.

For brevity and clarity, we summary the pipeline as five steps, as shown in Figure 2. First, we cluster customers into several initialized hyper-regions. Second, we generate an initial regional VRPs solution in individual hyper-regions via our elementary VRPs generator. Third, we utilize the rewriter to partition the merged graph to two sub-regions. Then our rewriter picks up two sub-regions via the attention mechanism and merges them into one hyper-region, and then generates the hyper-regional solution of the merged hyper-region. After that, we go back to the third step to re-partition the hyper-region into sub-regions in a loop. Through this process, the partition becomes more reasonable and the solution gets better and better. Finally after enough steps of rewriting we are able to reach a good solution.

## 3.1 REGION INITIALIZATION

Owning to a direct intuition that the spatially close customers are more likely to be scheduled within the same route in an optimal solution, an initial division based on the spatial locations is reasonable and benefits later convergence of the model's training. Therefore, we cluster customers according to their locations and divide the entire graph $G(V, E)$ into subgraphs $G_i$ as the initialization. We adopt K-means in this step for its effectiveness (Wagstaff et al., 2001). To accommodate to the generator model below, we set $K$ properly so that the number of customers in each cluster is around 100. Besides, in order to make full use of both customer locations and the depot location for clustering, we set the distance used for K-means as a linear combination of the Euclidean distance $d^E$ and the polar distance $d^P$, which is calculated using the cosine of the included angle $\theta$ in the polar coordinate system, whose center is at the depot and the axis is a fixed line. So the overall distance between customer $i, j$ is $d_{i,j} = d_{i,j}^E + \beta d_{i,j}^P$. In experiments we set $\beta$ to be 0.1. The detailed intuition analysis and ablation study on $\beta$ can also be found in Appendix.

## 3.2 GENERATING

For small-scale VRPs in merged hyper-regions, we follow the Attention Model (Kool et al., 2019) proposed for routing problems, such as CVRPs, TSPs and other variations. The attention model consists of an encoder to produce embeddings of all customers and a decoder to produce the strategy sequence. When a partial tour has been constructed, it cannot be changed, and the remaining problem is to find the following path from the last customer. The model benefits from the strong context representation ability of attention and the separate off-line inference stage of RL, and thus could generate high-quality solutions within an extremely short period.

Though we cannot directly apply it to large-scale problems as it is not trainable at large scale, it is a good choice to use it in small-scale local regions to achieve plausible results. To accelerate the training, we use the pre-trained model trained on two-center-clustered customer distribution and fine-tune it during training to suit the distribution in hyper-regions. Following Kool et al. (2019), we use the solution cost as rewards and REINFORCE algorithm (Williams, 1992) with a rollout baseline for the training of the generator.

### 3.3 REWRITING

The rewriter agent conducts a selecting-merging-repartitioning process to update region divisions. To be detailed, we consistently employ the attention mechanism to select region-pairs and merge them into hyper-regions. According to the newly generated solutions from each hyper-region, we further partition each hyper-region based on the inside routes into two regions.

We first generate the regional representations for region selection. We obtain local solutions from the generator in each region, in which different routes represented by sequences inside one region are sent as the input to an LSTM based network (Gers et al., 1999) to capture the sequential features. Further, we take the mean value of the outputs from LSTM of each route in a region and process it using a fully connected network to generate the $d_h$-dimensional feature $h_i$ for each region $G_i$,

$$h_i = \frac{1}{N_i} W^\tau (\sum_{j=1}^{N_i} \text{LSTM}(\tau_{i,j})) + b^\tau, \tag{1}$$

where $W^\tau$ and $b^\tau$ are shared weights and bias respectively, $N_i$ is the number of routes in region $G_i$, $\tau_{i,j} = (v_0, v_{i,j,1}, v_{i,j,2}, ..., v_{i,j,n_{i,j}}, v_0)$ is the $j$-th route starting and ending at the depot $v_0$ in region $G_i$, and $h_i$ as the representation of each region $G_i$ for further process. All regions share the same region encoder and its parameters.

With the well represented regional features, we first select region-pairs for further merging and repartition via the attention mechanism. The selection is processed in two steps to generate the region-pairs. First we compute the selection probability $p_{i,j}$ as

$$p_{i,j} = \text{softmax}_{j \in U_i}(h_i^T h_j), \tag{2}$$

So for each region $G_i$, $p_{i,j}$ sums up to one. To ease the training, for each region $G_i$ we restrict $U_i$ to the $K$ nearest regions to $G_i$ and set $p_{i,j} = 0$ for $j \notin U_i$. We intuitively set K=5,8,9 for N=500,1000,2000 cases, respectively, to allow for more neighbors considered as the problem size increases and spatial size of one region becomes smaller. Then we visit every region $G_i$ in a randomized order and select its pair region $G_j$ with probability $p_{i,j}$ to form the region-pair $(G_i, G_j)$. We drop region-pairs that intersect with previously chosen pairs. After this, we will obtain a bunch of region-pairs with no intersections with each other.

For each selected region-pair $(G_i, G_j)$, we now merge them together into a hyper-regional subgraph $G_{merge} = G_i + G_j$, and regenerate the corresponding hyper-regional VRPs solution $\pi^t_{merge}$ over the subgraph using the elementary VRPs generator. Since the hyper-regional solution considers the customer information more globally, it is comprehensible that the newly generated solution $\pi^t_{merge}$ is more likely to obtain better quality than $\pi^t_i + \pi^t_j$. If the new solution is better than the previous one, we accept the updating on the overall solution of all customers as follows,

$$\pi^{t+1} = \pi^t - \pi^t_i - \pi^t_j + \pi^t_{merge}. \tag{3}$$

To maintain the regional-scale consistency and prevent the dilation of any subgraph for further process, we repartition the merged hyper-region $G_{merge}$ back to regular sized region $G_{i'}$ and $G_{j'}$. Due to the same intuition in the initialization that an optimal solution is more likely to assign close customer into the same route, we calculate the spatial center of all customers within one route as its representations and use principal component analysis (PCA) to reduce the representations to 1 dimension, and then divide all routes by sides into two new regions $G_{i'}$ and $G_{j'}$ with similar number of customers.

**To summarize**, the rewriter takes a selecting-merging-repartitioning process to update region divisions. Such a step is called a rewriting step, or a rollout step. The rewriting is optimized with the performance improvement of the new division to the previous one.

### 3.4 Optimization via REINFORCE

In the above-mentioned steps, the repartition takes place on the route-level, which does not change the total distance cost. This means the total cost is only influenced by merging and recalculation. Hence we define the reward function for one region-pair $(G_i, G_j)$ as follows,

$$r = \mathcal{C}(\pi_i^t) + \mathcal{C}(\pi_j^t) - \mathcal{C}(\pi_{merge}^t). \tag{4}$$

It describes how the solution is improved in each round. Meanwhile, to guarantee an effective iteration, if the new generated solution is worse than the previous one, we reject the updated partition and solutions and continue the next rewriting step.

According to the reward, we optimize the model by gradient descent via REINFORCE method (Williams, 1992) with a baseline. To reduce the variance of the gradient propagation, we define the baseline $b$ as the running average of the rewards.

## 4 Performance Evaluation

**Dataset.** CVRP is a typical variant of VRP with capacity constrains. Generally, if models can work well on CVRP, it is easy to transfer to VRP by removing the capacity limitation. We follow the similar data generation method as the CVRP evaluation settings from previous works (Nazari et al., 2018; Kool et al., 2019; Chen & Tian, 2019) for consistency and fair comparison. The location of each customer and the depot $(x_i, y_i)$ is sampled uniformly from a unit square, which means both $x_i$ and $y_i$ are sampled uniformly from the interval $[0, 1]$. The demand $d_i$ of each customer is sampled from the discrete set $\{1, 2, ..., 9\}$, and the capacity of the vehicle is 50. For simplicity, we fix the depot at the central of the area. For random depot cases, we can simply shift and scale the data to make the depot central and customers in $[0, 1] \times [0, 1]$, and the result will still be good as shown in experiments below. The traveling distance between two customers or the depot are calculated directly using the inner Euclidean distance.

**Evaluation Protocols.** We consider three different evaluation settings for large-scale CVRP with customer amount $N = 500, 1000, 2000$ respectively. During the training process, we use a learning rate of 0.00003 for funetuning the generator and 0.001 for training the rewriter with SGD. At training, we run 10 rollout steps for each sample and randomly rotate the positions of all customers along the depot at each step to make training data in a near-i.i.d. distribution. We train for a total 10000 samples. At evaluation, we run for 100 rollout steps (i.e., rewriting steps) and take the solution at the last step as the final solution. Our method is implemented in Python via Pytorch framework, and the experiments are run on 4 Nvidia 2080Ti GPUs.

### 4.1 Performance Comparison with Baselines

In Table 1, we first present the performance of *Randomly Generated* whose routing solution is generated completely randomly as a background performance of the evaluated instances. Then, we compare our proposed RBG framework with other state-of-the-art baselines, including the heuristics and the RL ones. *Ant Colony* (Gambardella et al., 1999) and *LKH3* (Helsgaun, 2017) are two classic state-of-the-art heuristics, and *OR Tools* (Google, 2016) is the widely-used combinatorial optimization solver developed by Google. As for the state-of-the-art RL baselines, we select the Attention Model (Kool et al., 2019) which generates the solution tour by steps and the ReWriter (Chen & Tian, 2019) that rewrites the previous solutions for improvement. Specifically, we both evaluate the performance of Attention Model trained on only 100 (i.e., original design) and the according amount of customers. The time is computed as the average solution time for one instance.

As shown in Table 1, our RBG framework outperforms all other baselines on $N = 2000$, and is only slightly outperformed by LKH3 on $N = 500$ and $N = 1000$ with extremely narrow gaps with regards to the total distance, i.e., objection. It is interesting to find that the performance gap from LKH3, the state-of-the-art algorithms compared to both heuristics and RL based methods, keeps decreasing when the size of customers increases. When $N = 500$, the performance gap is 2.82%, and reduces to 0.99% when $N = 1000$. While RBG eventually outperforms LKH3 by 2.43% when the customer size keeps expanding to 2000. This demonstrates that our RBG framework not only generates high-quality solutions when the problem is at a large scale but also has a growing

Table 1: Overall performance comparison. The objection is the average total route length (less is better). Time is the average time to solve a single instance (less is better). AM_100 is the AM model trained on N=100 size, and AM is trained on the corresponding sizes and tested on the same size.

| | N = 500 | | N = 1000 | | N = 2000 | |
|---|---|---|---|---|---|---|
| | Obj. | Time | Obj. | Time | Obj. | Time |
| Randomly Generated | 273.49 | - | 546.12 | - | 1091.35 | - |
| Ant Colony (Gambardella et al., 1999) | 61.55 | 20min | 112.28 | 50min | 207.56 | 2h |
| LKH3 (Helsgaun, 2017) | 49.83 | 5min | 94.93 | 18min | 185.70 | 80min |
| OR Tools (Google, 2016) | 54.72 | 20s | 100.77 | 80s | 186.95 | 5min |
| ReWriter (Chen & Tian, 2019) | 60.67 | 33s | 108.82 | 37s | 198.76 | 8min |
| AM_100-greedy (Kool et al., 2019) | 55.48 | 50ms | 106.60 | 100ms | 218.68 | 200ms |
| AM_100-sampling (Kool et al., 2019) | 55.21 | 12s | 116.41 | 18s | 280.38 | 39s |
| AM-greedy (Kool et al., 2019) | 290.68 | 50ms | 637.17 | 100ms | 998.56 | 200ms |
| AM-sampling (Kool et al., 2019) | 269.31 | 12s | 603.70 | 18s | 954.95 | 39s |
| RBG | 51.24 | 7s | 95.87 | 15s | 181.19 | 30s |

advantage to other methods when the problem scale grows. This is of great practical value since maintaining solution quality is critical to many realistic industrial applications in which a vast number of customers may occur frequently Arnold et al. (2019).

We can further observe that learning RL-based solvers from scratch on large scale VRP cannot reach acceptable results as shown by AM-greedy and AM-sampling, the costs of which are close to randomly generated, because the training is unable to converge due to large action space and the delayed reward. In contrast, learning iteratively or learning to generate (ReWriter and Am-train_on_100) can improve the performance remarkably. Moreover, our proposed RBG can consistently achieve favourable objection gain compared to those two baselines. This further justifies the superiority of our heuristic RL design.

Apart from the solution quality, RBG also has remarkable inference efficiency. It only costs $30s$ to obtain the solutions to the instances with $N = 2000$. Compared to LKH3, the state-of-the-art method that has a close performance to RBG, our framework works about 100 times faster. A plausible reason is that LKH3 must infer a solution for each new problem from scratch, while our RL-based model can learn policy from the prior instances and generate solution promptly. Consequently, the ability of fast-responding to new instances makes our proposed RBG framework adaptive to situations where instances require real-time responses.

## 4.2 ANALYSIS OF REWRITING STRATEGY

In our framework, we propose two RL agents to perform different functions. Since previous literature has demonstrated the effectiveness of the generator RL, i.e., the Attention Model (Kool et al., 2019) in small scale VRPs, here in this section, we aim at discussing the influence of the rewriting RL.

Firstly, we showcase the importance of rewriter learning to the overall performance. To achieve this, we maintain the RBG framework without selecting regions from the RL agent. Instead, we randomly select two adjacent regions at each step directly. The comparison is shown in Figure 3. The y-axis is the ratio to the best performance minus one. We can observe that RL can both improve better and faster than random adjacent selection.

Besides, we analyze the extent and the frequency of repartition along with the performance improvement. Figure 4(a) shows the reallocation ratio by rollout steps, which is the ratio of customers reallocated after a repartition operation. It is calculated as $min(G_i - G_j \cap G_{i'}, G_i - G_i \cap G_{j'})/(G_i + G_j)$. Figure 4(b) shows the rate of repartition being accepted. Figure 4(c) shows the normalized traveling cost ratio along with rollout steps by the minimal cost. We found that the reallocation ratio and the partition update rate decreases significantly as long as the traveling cost decreases. The similar decreasing trend among them demonstrate that the improvement of the solution quality is highly related to the extent and frequency of rewriting. The rewriter finally tends to stop its rewriting operation when the solutions are close to optimal.

In conclusion, Rewriter shows its effectiveness on improving the solution quality during the instance inference.

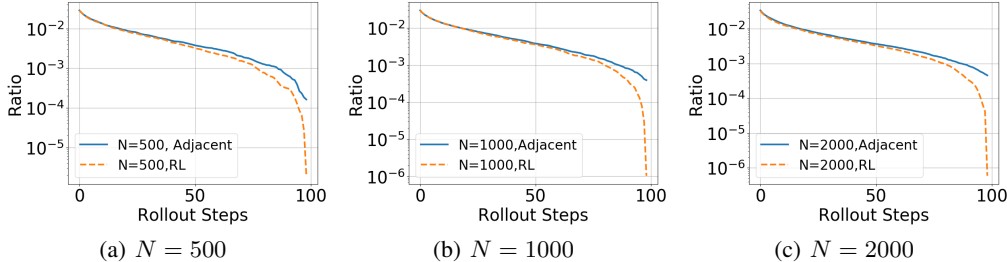

(a) $N = 500$       (b) $N = 1000$       (c) $N = 2000$

Figure 3: Comparison between RL and adjacent selection. The "Ratio" metric is the ratio of performance to the best performance obtained by these two strategies minus one. In order for clarity, it is plotted in log mode. A rollout step is a rewriting step, i.e., we divide-merge-regenerate the solution once. We run 100 rollout steps for each instance at evaluation.

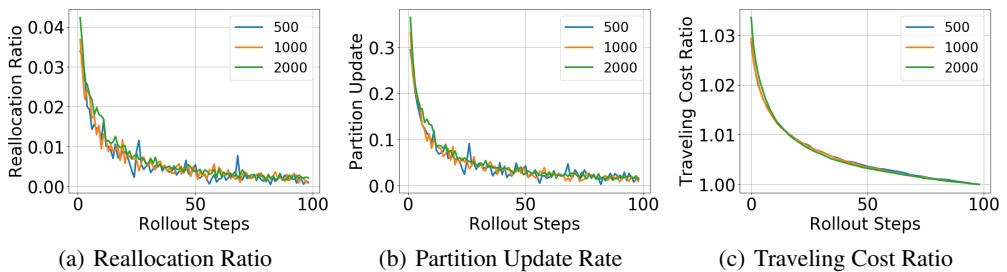

(a) Reallocation Ratio     (b) Partition Update Rate     (c) Traveling Cost Ratio

Figure 4: Statistics of the rewriter at different rollout steps.

## 4.3 ANALYSIS OF ROBUSTNESS TO DATA DISTRIBUTION

We conduct experiments to showcase the robustness of RBG to different data distributions. Following the same protocol of data generation in Fukasawa et al. (2006), we test RBG compared to LKH3 in three different scenarios in Fig 5: 1) **Random depot, random cluster**, in which the depot and the customers are all located randomly, 2) **Central depot, clustered customer**, whose customers distribute in several clusters, and 3) **Central depot, random-clustered customer**, in which half customers are generated randomly, and the others gather at several clusters, 4) **Central depot, random customer**, which is the same with our training setting. In all these scenarios,

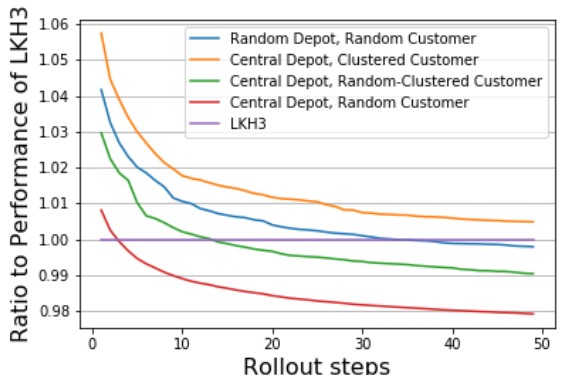

Figure 5: Performance on different data distributions.

the RBG is trained on the central depot and random customer data. For comparison, the performance of RBG in each scenario is normalized by the corresponding LKH3 traveling cost.

We find that our proposed RBG performs well across different data distributions and the decrease trends of the traveling cost are similar. Specifically, RBG outperforms LKH3 at about the 5-$th$, 10-$th$ and 50-$th$ step in the last three groups. Clustered customer is the only case where RBG is outperformed by LKH3. A plausible reason is that the clustering of the customer distributions make it easier for the LKH3 to find the close customers within the same cluster and assign them to the same route,, and thus the advantage of dividing customers into reasonable regions is weakened. However, RBG still obtains high-quality solutions with performance loss of no more than $0.1\%$. Detailed visualization is presented in Appendix.

## 5 DISCUSSION AND CONCLUSION

In this paper, we propose the Rewriting-by-Generating framework for solving large-scale routing problems. The framework generates regional routing solutions within each independent regions, and rewrites its previous performance by merging and repartitioning region pairs via an RL based agent. We demonstrate our framework generates high-quality solutions efficiently, and the performance advantage to other methods even increases when the problem scale expands.

Apart from solving VRPs efficiently as demonstrated by our experiments, our framework provides a way of enhancing the learning nature of RL mechanism with heuristic operations to solve a variety of large-scale optimization problem hierarchically. It needs minor effects to adopt our RBG for other problems which existing learning methods are only able to handle at small scale, and the key is to replace the generator by corresponding solvers. As the future work, we plan to generalize our framework to large-scale routing problems with different constraints as well as other NP-hard problems.

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

## A    Appendix

### A.1    Preliminary of VRP

In this paper, we evaluate our work on CVRP specifically. We present a complete mathematical formulation in this section. CVRP involves one depot and several customers with different demands to need. The task is to determine the routes for vehicles with limited capacity to traverse all the customers, and the target is minimize the total traveling cost.

Let $G(V, E)$ denote the graph consisting of depot and customers. Specially, $V = \{v_0, v_1, ..., v_i, ..., v_N\}$, where $v_0$ denotes the depot, and $v_i(1 \leq i \leq N)$ denotes the $i$-th customer with its location $(x_i, y_i)$ and demand $d_i$. For each pair of node $i$ and $j$, the edge $e_{i,j}$, or $E(v_i, v_j)$ in another manner represents the distance between node $i$ and $j$. The capacity of the vehicle is denoted by $D$, is the constraint to the maximum of its loaded shipment. Each vehicle also has a demand denoted by $d$. A vehicle must return to the depot $v_0$ to reload when all goods are delivered. We denote the solution of the subgraph $G_i$ as $\pi_k = \{v_{k,0}, v_{k,1}, ..., v_{k,N_k}\}$ and the corresponding cost as $\mathcal{C}(\pi_k) = \sum_{i=0}^{N_k} E(v_{k,i}, v_{k,i+1})$. The entire solution of all $K$ subgraphs is denoted by $\pi$.

With the above notations, we mathematically define VRP as follows,

$$\min \sum_{k=1} \mathcal{C}(\pi_k), \tag{5}$$

$$s.t. \quad \pi_1 \cup \pi_2 \cup \cdots \cup \pi_m \cdots \cup \pi_K = V, \tag{6}$$

$$\pi_m \cap \pi_n = \emptyset, m \neq n, \tag{7}$$

$$\sum_{i=p}^{q} d_i \leq D_m, v_0 \notin \pi_m^{p:q}, \forall q > p, q \in \mathbb{N}, p \in \mathbb{N} \tag{8}$$

where constraint (2) and (3) ensure all customers visited and only visited once with demands satisfied. (4) indicates the capacity constraint.

## A.2 ANALYSIS ON DIFFERENT INITIALIZATION STRATEGIES

Due to the intuition that the closely-distributed customers are more likely to appear in the same route in an optimal solution, we cluster the customers according to their locations as discussed in Section 3. However, the detailed spatial feature used for clustering may vary. Hence we analyze two different ways to measure the related distance between two customers $i$ and $j$: 1) **Euclidean distance**, $d_{i,j}^E$ and 2) the **polar distance**, $d_{i,j}^P$, which is calculated using the cosine of the included angle $\theta$ in the polar coordinate system, whose center is at the depot and the axis is a fixed line. The initial partition in Fig 1 shows an example of how customers are divided only according to their cosine distance.

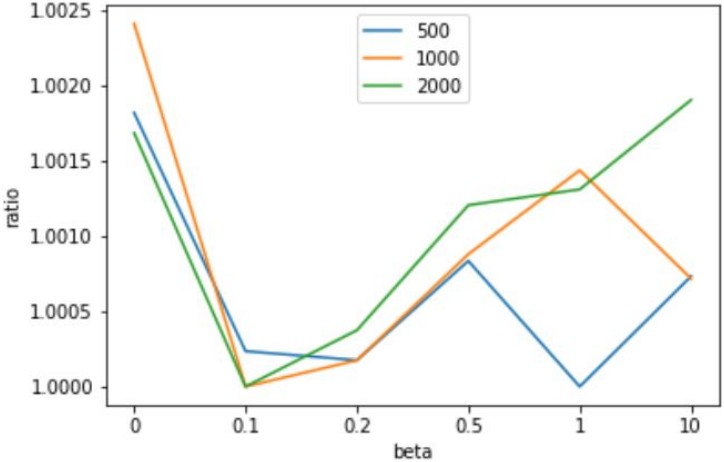

Figure 6: Ablation study on distance factor(beta) for clustering

To measure the influence of the two partition features, we use a combination of them, $d_{i,j}^E + \beta d_{i,j}^P$, to represent the distance between customer $i$ and $j$, where $\beta$ is a hyperparameter. The initialization is obtained via the classical K-means algorithm(Krishna & Murty, 1999; Alsabti et al., 1997) and the results are shown in Fig 6. The traveling costs are normalized by the minimal one. We find that the generated solutions has the best quality when $\beta = 0.1$. Euclidean distance is a straightforward feature to measure the closeness of customers, but the division that relies on it only may generate clusters that are far away from the depot. The corresponding route may suffer from a great distance cost to travel from the depot to the cluster, and then back to the depot. While the polar distance can prevent this shortage. An appropriate combination of them can benefit the RBG framework to obtain higher performance.

It is also remarkable to point out that the spatial feature combination has a low fluctuation range. The worst performance in every customer scales is no worse than $0.25\%$ than the optimal one. This shows the great robustness of RBG to different initialization strategies.

## A.3 VISUALIZATION OF INFERENCE

We present the visualization of the initial and final solutions in all three customer scales, as shown in Fig 8. It is worthy to mention that due to the initialization based on spatial features, the initial routes appear to be more gathering in space. Different routes can usually be separated by a clear boundary between them. However, in the final solutions which are rewritten and regenerated for steps by RBG, the routes tend to have more complicated intersections in space. This is because other factors, including the customer demands and the capacity are further considered by RBG and more reasonable divisions and routes are updated.

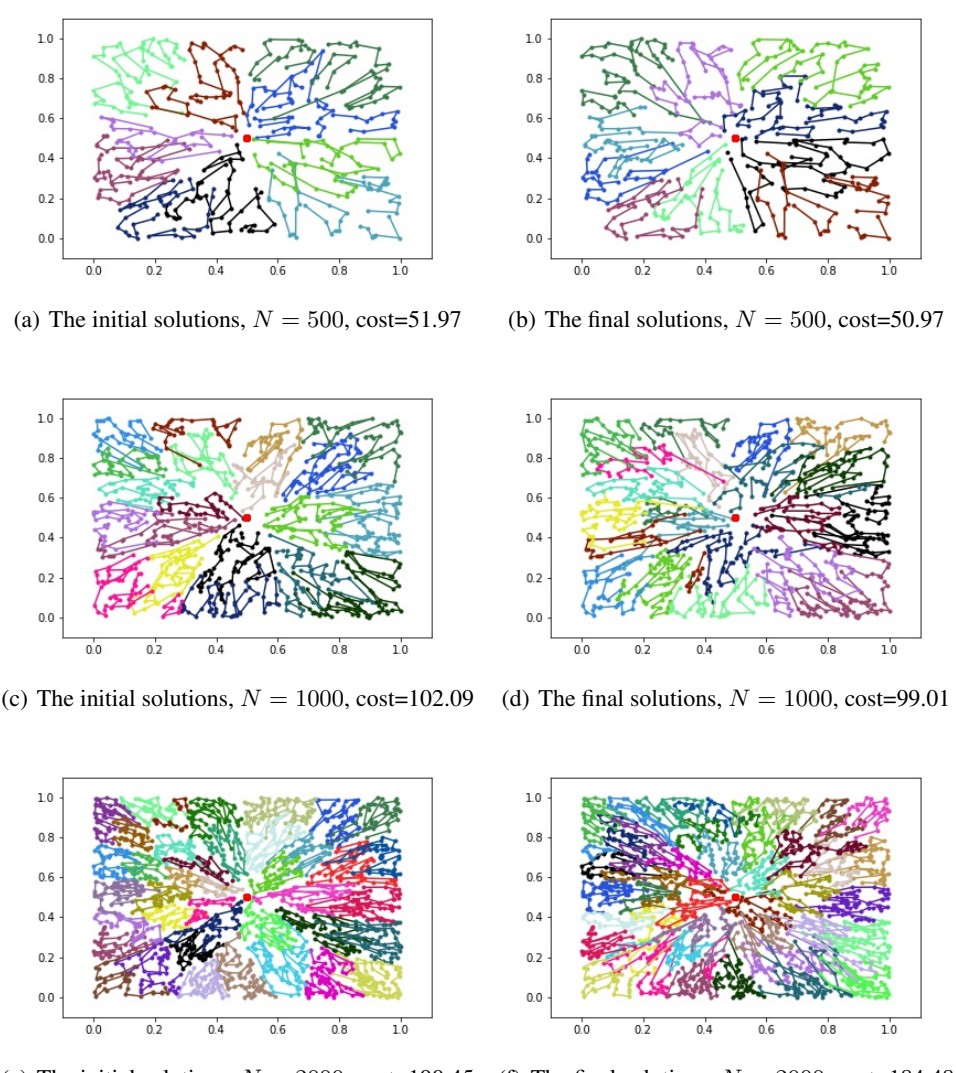

(a) The initial solutions, $N = 500$, cost=51.97    (b) The final solutions, $N = 500$, cost=50.97

(c) The initial solutions, $N = 1000$, cost=102.09    (d) The final solutions, $N = 1000$, cost=99.01

(e) The initial solutions, $N = 2000$, cost=190.45    (f) The final solutions, $N = 2000$, cost=184.48

Figure 7: Visualization on initial and final solutions of different scales. The big red point at the center represents the depot, and each blue point represents a customer. Routes with the same color are in the same region. For clarity in visualization, we omit the line segment from depot to the first customer and from the final customer back to depot for each route.

## A.4 VISUALIZATION OF SOLUTION ON DIFFERENT DATA DISTRIBUTION

## A.5 GENERALIZATION ABILITY

Considering that RL based model has generalization ability, that is to train and infer on different instances, we further design experiments to show how our RBG performs in this case. Specially, we train model for $N = 500$, 1000, and 2000, and compare the travel cost in Table 2. The similar results demonstrate the good generalization ability of our RBG.

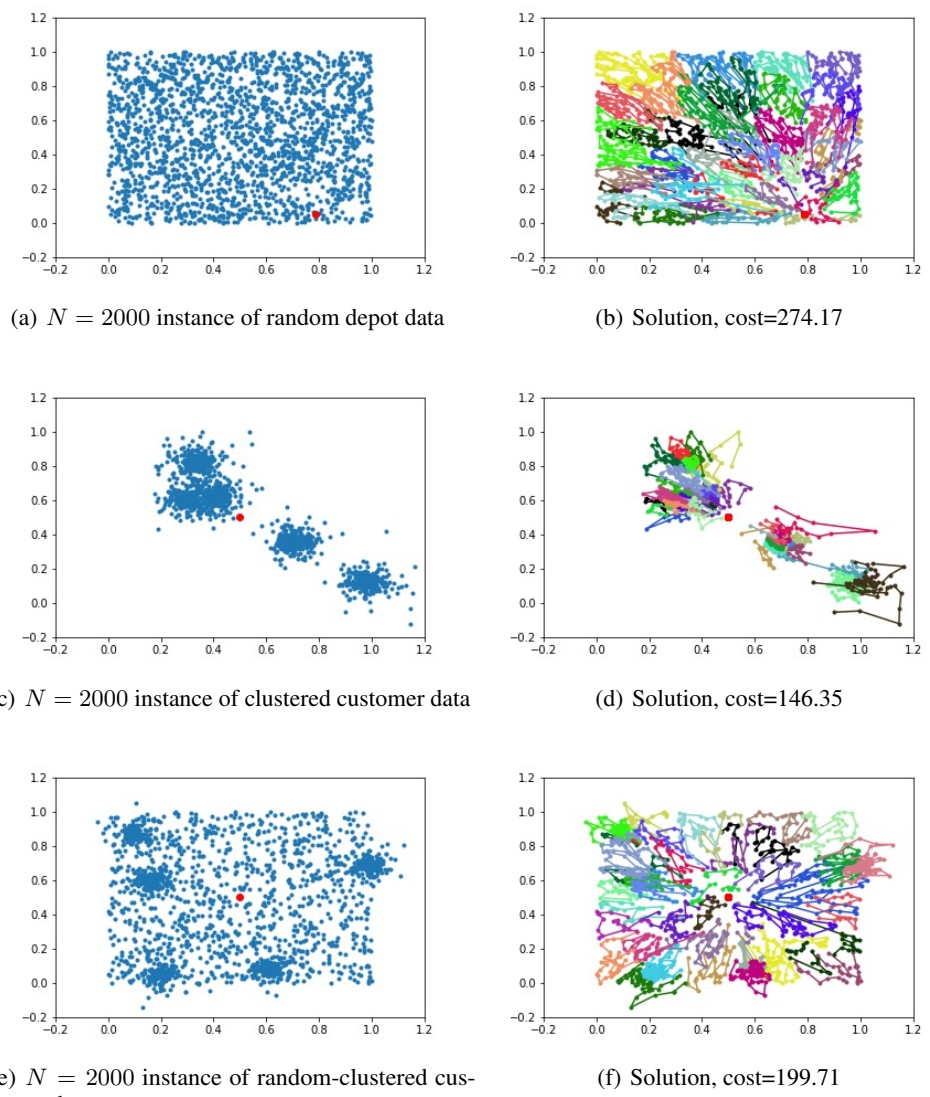

(a) $N = 2000$ instance of random depot data

(b) Solution, cost=274.17

(c) $N = 2000$ instance of clustered customer data

(d) Solution, cost=146.35

(e) $N = 2000$ instance of random-clustered customer data

(f) Solution, cost=199.71

Figure 8: Visualization on graphs and solutions of different data distributions. The big red point represents the depot, and each blue point represents a customer. Routes with the same color are in the same region. For clarity in visualization, we omit the line segment from depot to the first customer and from the final customer back to depot for each route.

Table 2: Generalization ability on scales

|          |      | Test on |       |        |
|----------|------|---------|-------|--------|
|          |      | 500     | 1000  | 2000   |
| Train on | 500  | 51.24   | 95.88 | 181.24 |
|          | 1000 | 51.22   | 95.87 | 181.28 |
|          | 2000 | 51.22   | 95.89 | 181.19 |

