# OpenReview forum: "Rewriting by Generating: Learn Heuristics for Large-scale Vehicle Routing Problems"
_ICLR.cc/2021/Conference — Reject_

### Official Review · AnonReviewer4 · 2020-10-28
**Improving neural VRP solving by improving smaller subproblems**

**Rating:** 6
**Confidence:** 3

**Review:**

# Summary

The paper proposes an extended framework for neural combinatorial optimization of the vehicle routing problem.
Whereas earlier works focused on the optimization of the whole solution at once, this method adds an abstract step to group and rewrite only parts of the route in a "divide-and-conquer" manner that allows the neural solver to focus on smaller subproblems and thereby avoid problems when scaling the problem to larger sizes.
The neural solver is based on the existing literature and the overall method is trained, as it is common in these papers, via Reinforce.
The method is well introduced and an experimental evaluation is performed on CVRP instances from 500-2000 customers, showing good scaling abilities of the presented method.

# Comments

The interest in solving combinatorial optimization problems using ML/RL is high and the method is a timely advancement in this area, since it addresses one of the main issues in earlier work, which is how to scale to larger instances.
The approach to repeatedly solve smaller subproblems of the large instance to achieve local improvements of the global search is an established technique and has been widely applied in the mathematical optimization community and operations research.
It is therefore reasonable to transfer the concept.
The application is straightforward and the main contribution of the paper is to apply this kind of divide-and-conquer and the selection of the partitions to merge and optimize using a LSTM model rather than a purely heuristic or random selection, which gives some small additional improvement.
The results show good improvements for large instances and the framework is general enough to be combined with new neural solvers, which will improve its adoption in the future.
The contribution is therefore valuable to the community since it introduces concepts that might not be known, but they are not particularly novel.
Nevertheless, the results are strong and the paper will be of interest to parts of the community.

Regarding the experiments, I'm not entirely clear how the baseline results are achieved.
Ant Colony Optimization is a surprising choice for a metaheuristic baseline and I'd guess that there should be better alternatives, but I'm not too familiar with the most recent literature here. Finding something more recent or performant would however strengthen the presentation of the results, even though I do not expect it to outperform the presented method.
Is OR-Tools run with a timeout or is it solved to optimality? Which CVRP model was used there?
In general, is there any information about the optimality gap of the found solutions?

In Sec. 4.2 (Analysis of Rewriting Strategy) I do not completely follow the metric used to compare both selection strategies.
Could the authors clarify this?
Also, the difference between both strategies are mostly small, which indicates that most of the performance benefits stem from the partitioning & rewriting in general and less from the learned selection strategy.
Since one of the strategies is random, did the authors test to pick a fixed heuristic strategy, e.g. pick the adjacent partition that has not been touched the longest or that had the biggest improvement before (or any other, there are several choices)?
Did you have a look at the strategy the LSTM converges to?

# Minor Comments

There are a couple of language problems in the paper and an additional round of proof-reading would be beneficial.
E.g. the "problem" in the title should be plural, the first sentence in the 2. paragraph of page 1 has some extra words ("those two lines"), and there are multiple other issues.

---

> ### Author Response · Authors · 2020-11-22
> **[Draft Update]Response to Reviewer4:**
>
> Thanks for your positive feedback, which helped us to improve our paper significantly, and we hope the following answers can be useful for addressing your concerns.
>
> $\textbf{Question 1:} $ The reason of comparison to Ant Colony baseline and possible alternatives.
>
> $\textbf{Response:} $ We select baselines from both traditional heuristics and recent learning based methods. As for the traditional heuristics, ant colony is selected as a native  baseline, while OR-tools and LKH3 stand for more integrated commercial solvers and state-of-the-art solutions. Thus, the ant colony here is rather only an academia representative, while LKH3 is the state-of-the-art meta-heuristic among all and could provide strong enough comparison. While, our contribution is solving the large scale VRP by learning methods. The major baselines are the recent proposed RL based approaches for VRP. In summary, the comparisons with up-to-date meta-heuristic solvers and RL methods as baselines are extensive for the performance evaluation.
>
>
> $\textbf{Question 2:} $ Whether runing with a timeout strategy, and what is the CVRP model selection in OR-Tools settings.
>
> $\textbf{Response:} $ In our experiment, OR-Tools is run to optimal, and its performance is worse than both LKH3 and our method. Thus if it is run with a timeout strategy, the performance will be worse. The fundamental CVRP model is used in running OR-tools. We use the code and default settings in the official guides:
> https://developers.google.com/optimization/routing/cvrp.
>
> $\textbf{Question 3:} $  Is there any information about the optimality gap of the found solutions?
>
> $\textbf{Response:} $ For large-scale VRP, since it is unfeasible to find the exact optimal solution, the optimality gap usually cannot be obtained. Thus, we compare the performance obtained by different heuristics or learning-based baselines in our experiments.
>
> $\textbf{Question 4: } $ Explanation of the metric used to compare both selection strategies.
>
> $\textbf{Response: } $  Figure 3 demonstrates the results of the two region-selection strategies (adjacent v.s. RL). We first compute the performance ratio of the current solution of every rollout step to the final optimal solution, whose value are larger than 1. We further reduce it by 1 to generate the final metric, and plot in the log scale. We utilize this metric to better showcase the performance improvement along the rollout steps and distinct the comparison between the two methods, as shown in Figure 3. We have clarified the above metric description in the caption of Figure 3.
>
> $\textbf{Question 5:} $ Whether the performance is gained from the partitioning&rewriting in general according to the close performance in Sec 4.2.
>
> $\textbf{Response:} $ We agree with that results in Sec 4.2 suggests the general rewriting process benefits for the performance gain compared to other methods, while the attention-selection design also provides further improvements. The major framework of RBG is the hierachical  structure that combines meta-heuristics and RL methods. It benifits from the meta-heuristic idea, integrated in the rewriter, to guarantee the high solution quality while the learning nature, especially the generator, guarantees the computation efficiency for inference. In the detailed rewriter design, either a fixed strategy or an RL agent can execute the heuristic operations. Sec 4.2 demonsrates that the RL agent is a better choice due to its performance improvement.
>
> $\textbf{Question 6:} $ Did the authors test to pick a fixed heuristic strategy, e.g. pick the adjacent partition that has not been touched the longest or that had the biggest improvement before (or any other, there are several choices)? Did you have a look at the strategy the LSTM converges to?
>
> $\textbf{Response:} $ Thanks for the constructive suggestions. The performance comparison analyzed in Sec 4.2 is a semi-fixed heuristic strategy as suggested. The selecting is not totally random by setting a hyperparameter K as the adjacent range, and the rewriter selects one region randomly from the K neighbors to operate. More deterministic heuristic strategy designs proposed by the reviewer are interesting suggestions, and we will upload the relative comparison once new results come out.
>
> As for the convergence of the RL based rewriter, it tends to select spatially close regions when the network converges, as shown in Figure 7. The visualization of the final solutions demonstrates that the network do not mix one region with another that is far away, but only generate close selections. The convergence results match with the simple fact that customer locations within a route should be close to reduce the traveling cost.
>
> $\textbf{Minor Commets:} $ Typo modifications.
>
> $\textbf{Response:} $ We appreciate you for pointing out the language problems.  We have improved the writting throughout our paper.

---

> > ### Comment · AnonReviewer4 · 2020-11-23
> > **Optimality of OR-Tools results**
> >
> > > In our experiment, OR-Tools is run to optimal, and its performance is worse than both LKH3 and our method. Thus if it is run with a timeout strategy, the performance will be worse.
> >
> > Yes, if you enforce a timeout the solution will usually be worse, that is correct.
> >
> > What I do not understand is how OR-Tools can be worse than LKH3 or your method if it is solved to optimality, since, by definition, optimality is the best possible result for the problem. I have doubts you are actually calculating the optimal solution.
> > Do you use a different set of constraints than in the other formulations? Or does OR-Tools have some default timeout internally?

---

> > > ### Author Response · Authors · 2020-11-24
> > > **Response**
> > >
> > > We appreciate for your timely updating on the review, and we apologize for our ambiguous expression of "optimal".  In our previous response, "OR-Tools is run to optimal" means that we report the best performance which OR-tools can reach under the settings, but not the real optimality among all possible solutions for the problem. In fact, OR-Tools can hardly achieve the real global optimality, according to Bello et al., "it is a common choice for general routing problems and provides a reasonable baseline between the simplicity of the most basic local search operators and the sophistication of the strongest solvers."
> > >
> > > We further exploited how OR-Tools handled its exploration process under its default settings as we apply. In the default searching strategy, the "automatic" local search, "lets the solver select the metaheuristic" directly. While for our evaluation it selects greedy-descent local search. OR-Tools can keep accepting improving local search neighbors without a timeout. However, a problem for such an algorithm is the presence of plateaus, where no local move will decrease the cost and that is when OR-Tools stop running. Thus, the final result is only a local minimum but not a global optimality. That is also why OR-Tools are worse than LKH3 (the state-of-the-art meta-heuristic so far according to Helsgaun et al. and Lu et al).
> > >
> > > For reference, LKH3, as a meta-heuristic method, outperforms OR-Tools on all scales in recent works, including on small scales (N=20,50,100) in both [2] and [3], and on large scales (N=500,1000,2000) in our experiment. However due to the rapid increase in solution space, both two methods face the challenge of exploration efficiency, while our proposed method could decompose the large scale problem and solve them via a hierarchical RL framework. This is the reason why RBG shows its superiority on both inference speed and solution quality when the problem scale becomes large enough.
> > >
> > > [1]Irwan Bello, Hieu Pham, Quoc V Le, Mohammad Norouzi, and Samy Bengio. Neural combinatorialoptimization with reinforcement learning. arXiv preprint arXiv:1611.09940, 2016.
> > >
> > > [2]Kool W, van Hoof H, Welling M. Attention, Learn to Solve Routing Problems! In International Conference on Learning Representations. 2019.
> > >
> > > [3]Lu H, Zhang X, Yang S. "A Learning-based Iterative Method for Solving Vehicle Routing Problems". In International Conference on Learning Representations. 2020.
> > >
> > > [4]Keld Helsgaun. An extension of the lin-kernighan-helsgaun tsp solver for constrained traveling salesman andvehicle routing problems. Roskilde: Roskilde University, 2017.

---

### Official Review · AnonReviewer2 · 2020-10-28
**Interesting hierarchical RL framework for large-scale VRPs**

**Rating:** 6
**Confidence:** 5

**Review:**

Summary
--------------
The paper presents a hierarchical reinforcement learning approach to solve large-scale vehicle routing problems (VRPs). A “rewriting agent” is responsible for dividing the customers into regions while a “generating agent” is responsible for computing the vehicle routes in each region, independently. The rewriting agent learns to score pairs of regions to be merged, using as a reward the reduction of VRP cost gained by the merge. The VRP costs are computed by the generating agent, which is based on the attention model of (Kool et al 2019), that is known to perform well for smaller scale VRPs.

Strong points
-------------------
1. The main contribution of the paper is the novel rewriting process that allows to decompose the problem into smaller subproblems, that can then be tackled with state-of-the-art methods.
2. The numerical experiments show that this approach allows to efficiently solve large problems (2000 nodes).

Weak points
------------------
3. Many confusions in the text, in particular regarding the related works (see Feedback to improve the paper)
4. To validate the hierarchical framework and the rewriting process, it would have been great to apply it to other large scale combinatorial problems. For instance, the multiple vehicle routing problem, where the rewriter would be responsible of assigning the customers to the vehicles.
5. The authors do not mention whether they will release their code.

Recommendation
-------------------------
I would vote for accept. To me the hierarchical framework and the rewriting process are novel in this context and can potentially be applied in other problems.


Questions to authors
-----------------------------
6. “large-scale VRP is an *unexplored* and challenging problem”: VRP is one of the most studied problems in Operations Research. Among other aspects, there are of course works that aim at solving large scale problems. What do you mean by unexplored?
7. In Section 3.3, after the selection and merging phases, the hyper-regions are again split into 2  “regular sized” regions, while maintaining the routes computed in the hyper-region. Is this always guaranteed to be feasible? Say you end up with 3 routes, of 10 customers each, in the hyper-region. How to split them in this case?
8. “These heuristics usually have a time complexity of O(n 2 log n 2 )” can you share a reference for this claim?
9. Sec 3.3 “we restrict Gj to the K nearest regions to Gi”. How is K chosen?
10. Table 1: The results reported for AM-sampling vs AM-train-on-100 are a bit surprising to me. The paper (Kool et al 2019) reported the best performance when the model was trained and tested on the same size. Do you have an idea of why it’s not the case here?
11. Regarding AM-train-on-100, was sampling or greedy rollout used?
12. Table 2: are these results averages over a number of runs? Do you have an intuition about why the values are so close in all cases?
13. Figure 3: what are the "rollout steps" here exactly? Do you have an explanation for the rapid drop of the ratio right before 100?
14. Figures 7 and 8: Why are the routes not starting and ending at the depot in red?

Feedback to help improve the paper
---------------------------------------------------
15. In the introduction, “since pre-defined rules are not suitable for various cases of customer distribution, those methods have poor generalization ability”. It does not make sense to talk about generalization ability for non-learning based methods.
16. The “exploration space of large-scale VRP” / “exploration complexity” is confusing. I think you mean solution space instead of exploration space.
17. There are confusions at several points regarding heuristics vs learning algorithms: “....can be divided into heuristics and reinforcement learning (RL)”. RL-based approaches are also computing heuristics to solve the VRP, in the sense that they are not computing exact solution of the problem. So I guess here by heuristics, the authors mean non-learning based heuristics.
18. The Ant Colony baseline of 1999 does not look competitive at all with the other standard Operations Research heuristics for VRP (here LKH3 and OR Tools) so I don’t see the point in reporting its results (Table 1).
19. Sec 4.1 “…many realistic industrial applications in which a vast number of customers may occur frequently.” Do you have any reference for that? I would guess that when the number of customers is really large, it’s more likely to resort to the multiple vehicle problem.
20. Sec 4.3 “…this is the rare case in real-world.” Do you have a reference? Intuitively I tend to disagree, customers are likely to be grouped within cities for instance.
21. For a variety of VRP instances, you could use CVRPlib: http://vrp.atd-lab.inf.puc-rio.br/index.php/en/

---

> ### Author Response · Authors · 2020-11-22
> **[Draft Update]Response to Reviewer2: Part I**
>
> Thank you for your valuable time and comprehensive comments, which can help us improve our paper significantly. Due to the word limit of the comment, we will split our response into several parts.
>
> $\textbf{Weakness 4: } $Reason of not addressing other combinatorial problems for validation.
>
> $\textbf{Response:}$ We propose an effective RL based solution to tackle the large-scale challenge in solving VRP, which follows the same problem setting concentration as the recent work in ICLR( Lu et al., 2020) where only the fundamental CVRP is selected as the problem background. Due to the page limit we did not address too many detailed considerations on VRP, including the multi-vehicle setting as the reviewer stated. Meanwhile, our model is capable to be applied to other large scale combinatorial problems with more constraints and considerations. For instance, in the vehicle routing problem with time window(VRPTW), the cooperation between vehicles and the individual time windows are additional constraints. The method for such an adaptation is to replace the generator by the one that includes these new constraints, such as the small-scale RL based VRPTW solver proposed in(Zhang et al., 2020) recently, while the entire hierarchical structure remains the same. Thanks for this comment and we are going to explore the  generalizability to not only the variants of VRP, but also other combinatorial optimizations as our future work.
>
> [1]Lu H, Zhang X, Yang S. A Learning-based Iterative Method for Solving Vehicle Routing Problems[C]. International Conference on Learning Representations. 2019.
>
> [2]Zhang, Ke, et al. "Multi-Vehicle Routing Problems with Soft Time Windows: A Multi-Agent Reinforcement Learning Approach." arXiv preprint arXiv:2002.05513 (2020).
>
> $\textbf{Weakness 5:} $ Whether the authors will release the code.
>
> $\textbf{Response:}$ We have already included the codes in the supplementary material with pdf submission.  https://openreview.net/attachment?id=xxWl2oEvP2h&name=supplementary_material. $We also released our code in the Github Repo: https://github.com/RBG4VRP/Rewriting-By-Generating and will keep updating.
>
> ================================================================
>
> $\textbf{Question 6:} $ Explanation on the expression word  'unexplored'.
>
> $\textbf{Response:} $ We agree with the reviewer that  "unexplored" is not a rigorous statement. We intended to mean that large-scale VRP is unexplored for learning-based methods originally. We have clarified it in the paper, "Therefore, providing effective and efficient solutions for large-scale VRP is a challenging problem."
>
> $\textbf{Question 7:} $ How to guarantee the feasibility of hyper-region splitting?
>
> $\textbf{Response:} $ We divide the hyper-region into regular-sized regions according to the features of 'routes' instead of 'customers'. Therefore, maintaining the routes computed in the hyper-region can always be guaranteed. The detail is as follows.
>
> In the example provided by the reviewer, if there are route A, B, C in a hyper region, we will first get the geographic centre of all customers in A, B and C separately. Based on the spatial proximity, we can assign A with its customers as region G_1, and combine B, C into region G_2. Finally region G_1 and region G_2 have 10 and 20 customers, respectively. We thus clarify here that although we can always maintain the routes in the hyper-region, the "regular size" condition, i.e., close number of customers, is only a closed approximation but not a strict constraint.
>
> "Regular sized" regions are optimal, but extreme cases may happen, e.g., uneven customer distribution after splitting as in the example above. But the generator is generalizable on a range of problem sizes, thus it can still handle the scale after merging two non-standard regular sized regions. In addition, as we observed in our experiments, after several merge-and-split steps with other regions, the number of customers in every region is tending to be close.
>
> $\textbf{Question 8:} $ Reference of “These heuristics usually have a time complexity of O(n 2 log n 2 )” .
>
> $\textbf{Response:} $ After reviewing the related literature, we find that the time-complexity statement is not accurate enough. We appreciate the reviewer for pointing it out. The time complexity of O(n 2 log n 2 ) actually  denotes the complexity of VRPTW solved by meta-heuristics(El-Sherbeny, 2010). In general, solving CVRP by heuristics have a polynomial complexity, and may vary according to the detailed method. We have removed the statement in the paper.
>
> [1]El-Sherbeny N A. Vehicle routing with time windows: An overview of exact, heuristic and metaheuristic methods[J]. Journal of King Saud University-Science, 2010, 22(3): 123-131.

---

> ### Author Response · Authors · 2020-11-22
> **[Draft Update]Response to Reviewer2: Part II**
>
> This is the following response after $\textbf{[Draft Update\]Response to Reviewer2\: Part I}$ .
>
> $\textbf{Question 9:} $ How is K chosen in the expression “We restrict Gj to the K nearest regions to Gi”.
>
> $\textbf{Response: } $ We set K emperically as a hyperparameter. The intuition is that the solution is more likely to improve when two "near" regions are merged. When the problem size increases, the regions will have more neighbor regions under a global scope. Therefore, for N=500,1000,2000, we choose K to be 5,8,9 increasingly. We have made the details more clear in Sec3.3.
>
> $\textbf{Question 10:} $ Why AM-train-on-100 have better performance compared to  AM-sampling?
>
> $\textbf{Response:} $ On small-scale problems, it is true that training on the same size as test data will result in the best performance as shown by AM-sampling as the previous work presented. However, this characteristic does not remain in large-scale problems as the exponential growth in complexity. When the scale is large, directly training on such data suffers from terribly delayed reward and the enormous searching space, which makes it hard to be trained and find the close-optimal solution (This is also part of the motivation of our work). On the other hand, the model trained on small size may learn easier, while the knowledge it learned is generalizable(e.g., choosing the near customers from the current position may be better than choosing the far-away customers),  thus it can perform relatively good compared to AM-sampling on larger graphs. Note that we have renamed AM-train-on-100 to AM_100-greedy for better formatting.
>
> $\textbf{Question 11:} $ Regarding AM-train-on-100, was sampling or greedy rollout used?
>
> $\textbf{Response:} $ AM-train-on-100 is greedy, we have clarified it in the paper by renaming it to AM_100-greedy. We appreciate the reviewer for pointing it out.
>
> $\textbf{Question 12:} $ Whether the results averages over a number of runs in Table 2 and any intuition about why the values are so close in all cases.
>
> $\textbf{Response:} $ The result in Table 2 is a single run with the same random seed. The result suggests that our framework has a good ability of generalization, and knowledge on different scales is transferable to another. More experiments to generate the averages over a number of runs can be a great augmentation to the evaluation results. We appreciate for the reviewer's careful review and we are supplementing  experiments accordingly.
>
> $\textbf{Question 13:}$ Figure 3: what are the "rollout steps" here exactly? Do you have an explanation for the rapid drop of the ratio right before 100?
>
> $\textbf{Response:}$  A "rollout step" is a rewriting step, i.e., we split-merge-regenerate the solution once. We run 100 rollout steps for each instance at evaluation.
>
> For the second question, we have clarified the metric description in the caption of Figure3, where we compare the result of the two region-selection strategies (adjacent v.s. RL). We first compute the performance ratio of the current solution of every rollout step to the final optimal solution, whose value is larger than 1. We further reduce it by 1 to generate the final metric for comparison. For a better comparison we plot the figure by computing the log value.  Since the performance is tending to be the optimal when the rewriting gradually ends, the metric is approaching to zero and the log value thus drops rapidly.
>
> $\textbf{Question 14:} $ Figures 7 and 8: Why are the routes not starting and ending at the depot in red?
>
> $\textbf{Response:}$ Every route starts and ends at the depot indeed. In these figures, we omit these two line segment, one starting and another one ending at the depot, for clarity in visualization, referring to Kool et al, 2019. The visualization will be too vague if they are plotted. We have clarified it in the description of Figure7 and Figure8 in the paper.
>
> ================================================================

---

> ### Author Response · Authors · 2020-11-22
> **[Draft Update]Response to Reviewer2: Part III**
>
> This is the following response after $\textbf{[Draft Update\]Response to Reviewer2\: Part II}$ .
>
> $\textbf{Weakness 3 and Feedback 15,16,17:} $ Confusing expressions.
>
> $\textbf{Response:} $ We agree with the reviewer on that 1) it does not make sense to discuss the generalization ability for non-learning based methods, 2) 'exploration space' in RL is indeed the 'solution space' in the studied problem, 3) when we compare heuristics with RL, heuristics mean non-learning ones.
>
> To make our paper more precise, we have revised it according to your comments, and for your convenience, we quote the corresponding context as follows,
>
> Feedback 15: we deleted this "since pre-defined rules are not suitable for various cases of customer distribution, those methods fail to learn from previous cases and need to generate solutions from scratch for each new case."
>
> Feedback 16: "More specifically, the solution space of a large-scale VRP with $1000$ customers", "are difficult to solve large-scale VRP due to either high computation cost or the explosive solution space that results in model divergence."
>
> Feedback17: "Current algorithms proposed for routing problems can be divided into traditional non-learning based heuristics and reinforcement learning (RL) based models."
>
> $\textbf{Feedback 18:} $ The reason of comparison to Ant Colony baseline of 1999.
>
> $\textbf{Response:} $ We select baselines from both traditional heuristics and recent learning based methods. As for the non-learning heuristics, the ant colony is selected as a native well-known academia baseline proved by its 1227 citations up to now, while OR-tools and LKH3 stand for more integrated commercial solvers. The chosen of ant colony here is rather only an academia representative, while LKH3 is the state-of-the-art meta-heuristic among all and could provide strong enough baseline comparison. Besides, the motivation of our proposed framework is to tackle the large scale problem for learning methods. The major comparison of our experiments is with the recent RL based approaches for VRP. Providing comparisons with up-to-date methods from both two techniques is strong enough for the experiment comparison.
>
> $\textbf{Feedback 19: } $Reference for the expression “…many realistic industrial applications in which a vast number of customers may occur frequently.”
>
> $\textbf{Response:} $ Arnold et al., for example, claimed that "even though some applications like waste collection and parcel distribution can require tens of thousand of customers to visit, almost all research thus far has been targeted at solving problems of no more than a few hundred customers." Here we also aim to point out that customer size may explode into much larger scales than the algorithm could estimated, while our framework can still be pratical.  We have added the reference in Sec 4.1. Besides, CVRP, the setting we focus on, does not limit the number of vehicles as generated routes can be assigned to multiple vehicles finally. In Arnold's paper, a general VRP solver was proposed without considering multiple vehicles as well. In a nutshell, to solve large-scale VRPs, the multiple vehicle solution is one but not the only method.
>
> [1]Arnold F, Gendreau M, Sörensen K. Efficiently solving very large-scale routing problems[J]. Computers & Operations Research, 2019, 107: 32-42.
>
> $\textbf{Feedback 20:} $ Misunderstanding Claim on “…this is the rare case in real-world.”
>
> $\textbf{Response:} $ By looking into more references, we realize that our claim 'this is the rare case in real-world' is imprecise. We have removed it from the manuscript. In this section when comparing RBG with LKH3, we aim to show that LKH3 only outperforms our framework in 'Clustered customers' as illustrated in Figure 5, while we have superiority for all other ones.
>
> $\textbf{Feedback 21:} $ The recommendation of instances from CVRPlib.
>
> $\textbf{Response:} $ Thank the reviewer for the instance recommendation. The instances we use in the paper is generated in the same manner with precious literature for a fair comparison. We will consider using the Lib in your suggestion in our future work.

---

### Official Review · AnonReviewer3 · 2020-10-28
**Learning heuristics for CVRP, though important, is a very niche problem.  The proposed solution is very similar to what is already adopted by many of the meta-heuristics used by the OR solvers.**

**Rating:** 4
**Confidence:** 5

**Review:**

The authors propose a learning approach for developing heuristics to solve the capacitated vehicle routing problem (CVRP).  The paper is clearly written and easy to follow.

My main concern is related to generalizability of the proposed solutions.  Learning to optimize is an important emerging area. What is missing in this work is what the generalizability of the proposed methods to other combinatorial problems is. Specifically, why did the authors choose VRP?  VRP, though important, is a very specific problem with a lot of very good solutions using simple combinatorial techniques.  The other concern is related to the proposed techniques.  The approaches suggested by the authors, viz., partitioning the input, and rewriting the basic VRP solution by merging regions and recomputing routes, are also adopted by the meta-heuristics developed and used in the commercial OR solvers.

Another big drawback with the current model is that it does not address many constraints/considerations in real VRP applications that typical solvers address - different costs per vehicle, cost of missed shipment, route limits, dimension limits, alternate visits, etc.  Even the more realistic fixed number of vehicles case is not considered in the proposed solution.

---

> ### Author Response · Authors · 2020-11-22
> **[Draft Update] Response to Reviewer3:**
>
> We appreciate these valuable comments from you, which are helpful for us to improve our paper. We respond to your question as follows,
>
> $\textbf{Question 1:} $ The reason to choose VRP as the target problem and the generalizability to other combinatorial optimizations.
>
> $\textbf{Response:} $ Existing heuristic approaches for VRP face the major shortcomings of computational efficiency. In order to solve this problem, learning based methods has aroused much  attention recently, including many works in ICLR and NeurIPS (Kool et al., 2019; Nazari et al.,2018; Lu et al., 2020), which could generate VRP solution within seconds while the heuristic have to take hours. However,  all these methods only perform well when trained on usually no more than 100 customers, which cannot handle large-scale instances, i.e., thousands of customers, due to the unstable training caused by the exponentially expanding exploration space. Thus, it is important to develop a learning based solution to solve large-scale VRP, following the same problem setup proposed by the above works. This is the major reason that we choose VRP as the target problem.
> Moreover, our proposed solution can also be generalized as a framework for other specific combinatorial optimizations. For instance, to solve the large-scale KnapSack problem, the large amount of items and the total weight constraint can be clustered into groups. A global RL based rewriter with the current design is responsible to merge-repartition the regions while the the generator keep generating local solutions. The rewriter and the generator are trained under a hierarchical framework, which forms RBG. We have supplemented the discussion on generalizability in Sec 5.
>
> $\textbf{Question 2:} $ Technical novelty of the proposed framework compared to meta-heuristics.
>
> $\textbf{Response:} $ The motivation of this paper is to propose a learning based framework to solve the large-scale VRP. The high solution quality of previous meta-heuristics and the computation efficiency of reinforcement learning motivate us to combine the ideas from both fields of Operations Research and machine learning together. Thus, our designed rewriter agent is inspried by the idea of merging-repartitioning operations, and the related technical contribution is to conduct operations for exploration via a learning-based agent instead of a fixed or handcrafted heuristic strategy. The strength of such operations is able to construct an effective exploration process in the enormous searching space, while the hierarchical reinforcement learning design can guarantee the computation efficiency from the prospect of fast solution generation on test instances that have similar distribution with training ones.  Moreover, the combination of two techniques can also inspire solution to solve other combinatorial optimization problems. We have added the related discussion in the fourth paragraph of Sec 3.
>
> $\textbf{Question 3.} $ The reason of not addressing constraints/considerations in real VRP applications in the current model.
>
> $\textbf{Response:} $ The main contribution of this paper is proposing an effective RL based learning solution to solve the large-scale VRP, which follows the same problem setup as the recent work in ICLR (Lu et al., 2020) where only the basic CVRP is selected as the problem setup. Thus, we did not address the detailed constraints/considerations in real VRP applications, since a realistic solver with details is not our first goal. Meanwhile, our model have the ability to consider more detailed real-life constraints with minor respective modification on the generator. For example, to solve the vehicle routing problem with time window(VRPTW) with the consideration of the time window constraint and the cooperations between different vehicles, we only need to replace the generator by the one that includes these new constraints, for example, the RL based VRPTW solver (Zhang et al., 2020). The entire hierarchical structure remains the same. More generally, new constraints/considerations can be handled by designing adaptive generators for small-scale problems alone, and the supplemented discussion on these considerations is added in Sec 5.
>
> [1]Kool W, van Hoof H, Welling M. Attention, Learn to Solve Routing Problems! In International Conference on Learning Representations. 2018.
>
> [2]Nazari, Mohammadreza, et al. "Reinforcement learning for solving the vehicle routing problem." In Advances in Neural Information Processing Systems. 2018.
>
> [3]Lu H, Zhang X, Yang S. "A Learning-based Iterative Method for Solving Vehicle Routing Problems". In International Conference on Learning Representations. 2019.
>
> [4]Zhang, Ke, et al. "Multi-Vehicle Routing Problems with Soft Time Windows: A Multi-Agent Reinforcement Learning Approach." arXiv preprint arXiv:2002.05513 (2020).

---

### Official Review · AnonReviewer1 · 2020-10-30
**It proposes a hierarchical RL algorithm to solve large-scale VRP problems.**

**Rating:** 7
**Confidence:** 5

**Review:**

An RL based method, called Rewriting-by Generating (RBG), is proposed to solve large-scale VRPs. It borrows the idea of the hierarchical RL agent, which consists of two parts: "Generator" and "Rewriter". In the generation process, the graph is divided into several sections and in each section, an RL algorithm runs to get the best route. Then, the rewriter gets the solution of all generators and tries to connect them together with the goal of globalizing them with a smaller route. To this end, the rewriter merges each of two sub-problems together and then divides it into another two sub-problem and solves each again. Doing this helps decrease the route length. This diving and merging is learned by an RL agent (think of the outer agent in the hierarchical RL) so that the rewriter learns when and how to do this. The rewriter uses the attention mechanism to choose two parts of the merged routes, and then get a new solution for each part using the inner-agent. To get the initial sub-problems, K-mean clustering is used to get sub-problems of about 100 nodes. In the evaluations, CVRP of size 500, 1000, and 2000 are considered. The results are compared to LKH3 and google OR-Tools, along with RL algorithms. LKH3 slightly outperforms RGB in terms of the tour length in problems of 500 and 1000 nodes, though it takes a longer time to get the solution.
Question-1: How did you define a route $\tau_{i,j}$? How many of them can be there?

minor typo:
Then we then visit

---

> ### Author Response · Authors · 2020-11-22
> **[Draft Update]Response to Reviewer1: Notation Redefined, Typo Corrected**
>
> We appreciate your careful review and we have revised our paper according to your comments.
>
> $\textbf{Question 1:} $ How did you define a route $\tau_{i,j}$? How many of them can be there?
>
> $\textbf{Response:} $ In Eq.(1), $\tau_{i,j}$ denotes the $j$-th route in the $i$-th region. "A route" here is a sequence of customers starting from and going back to the depot, represented by $\tau=(v_0, v_1, ..., v_n, v_0)$, where $v_0$ is the depot, and $v_k=(x_k, y_k, d_k)$ is the $k$-th customer in this route. $(x_k, y_k)$ is the position of the customer, and $d_k$ is the corresponding demand. We use LSTM to capture the sequential pattern. To clarify the notation, we have improved the detailed definition in Sec. 3.3.
> The number of routes in one region depends on the current solutionto the local instance from the generator. The typical scale of one single region includes around 50 customers and 5 routes with each route containing around 10 customers. The scale is rather small since the generator is trained on no more than 100 customers to guarantee the trained ability.

---

### Decision · Program_Chairs · 2021-01-07
**Final Decision**

**Decision:**

Reject

**Comment:**

The authors propose an RL-based approach, “Rewriting-by Generating (RBG)”, to solve large-scale capacitated vehicle routing problems (CVRPs): such problems are NP-hard in general and are ubiquitous. The RL agent consists of a "Generator" and "Rewriter". In generation, the graph is sub-divided into several regions and in each region, an RL algorithm runs to get the best (or near-optimal) route. The rewriter then patches these near-optimal sub-solutions together using “hierarchical RL”.
The paper is generally well-written.

One main concern is related to generalizability: the authors respond that their approach can work for other NP-hard combinatorial-optimization problems such as knapsack. The authors are encouraged to do a systematic study of several such (related) problems where their approach can work. It was also a concern that the overall approach of partitioning the input instance and rewriting the CVRP solution by merging regions and recomputing routes, is also employed by commercial OR solvers. The authors are encouraged to do a careful comparison (and perhaps melding) with such available solvers, to get a hybrid “OR + ML” improvement. It is also suggested that the authors include several different constraints from real-world VRP (e.g., heterogeneous vehicle costs, costs of missed shipment, route limits, upper-bounded number of vehicles etc.).